# Role of Urinary Kidney Stress Biomarkers for Early Recognition of Subclinical Acute Kidney Injury in Critically Ill COVID-19 Patients

**DOI:** 10.3390/biom12020275

**Published:** 2022-02-08

**Authors:** Gustavo Casas-Aparicio, Claudia Alvarado-de la Barrera, David Escamilla-Illescas, Isabel León-Rodríguez, Perla Mariana Del Río-Estrada, Natalia Calderón-Dávila, Mauricio González-Navarro, Rossana Olmedo-Ocampo, Manuel Castillejos-López, Liliana Figueroa-Hernández, Amy Peralta-Prado, Yara Luna-Villalobos, Elvira Piten-Isidro, Paola Fernández-Campos, Santiago Ávila-Ríos

**Affiliations:** 1Centro de Investigación en Enfermedades Infecciosas, Instituto Nacional de Enfermedades Respiratorias Ismael Cosío Villegas, Calzada de Tlalpan 4502, Mexico City 14080, Mexico; claudia.alvarado@cieni.org.mx (C.A.-d.l.B.); isabel.leon@cieni.org.mx (I.L.-R.); perla.delrio@cieni.org.mx (P.M.D.R.-E.); mauricio.gonzalez@cieni.org.mx (M.G.-N.); amy.peralta@cieni.org.mx (A.P.-P.); andrea.luna@cieni.org.mx (Y.L.-V.); elvira.piten@cieni.org.mx (E.P.-I.); santiago.avila@cieni.org.mx (S.Á.-R.); 2Subdirección de Atención Médica, Instituto Nacional de Enfermedades Respiratorias Ismael Cosío Villegas, Calzada de Tlalpan 4502, Mexico City 14080, Mexico; davidei.md@gmail.com (D.E.-I.); rossanaoo@gmail.com (R.O.-O.); 3Departamento de Enseñanza, Instituto Nacional de Enfermedades Respiratorias Ismael Cosío Villegas, Calzada de Tlalpan 4502, Mexico City 14080, Mexico; nataliacd_26@hotmail.com (N.C.-D.); paolafer011@gmail.com (P.F.-C.); 4Departamento de Epidemiología, Instituto Nacional de Enfermedades Respiratorias Ismael Cosío Villegas, Calzada de Tlalpan 4502, Mexico City 14080, Mexico; mcastillejos@gmail.com; 5Laboratorio Clínico, Instituto Nacional de Enfermedades Respiratorias Ismael Cosío Villegas, Calzada de Tlalpan 4502, Mexico City 14080, Mexico; ilifh@hotmail.com

**Keywords:** acute kidney injury, COVID-19, neutrophil gelatinase-associated lipocalin, tissue inhibitor of metalloproteinases-2, insulin like growth factor binding protein 7, urinary kidney stress biomarkers

## Abstract

A high proportion of critically ill patients with COVID-19 develop acute kidney injury (AKI) and die. The early recognition of subclinical AKI could contribute to AKI prevention. Therefore, this study was aimed at exploring the role of the urinary biomarkers NGAL and [TIMP-2] × [IGFBP7] for the early detection of AKI in this population. This prospective, longitudinal cohort study included critically ill COVID-19 patients without AKI at study entry. Urine samples were collected on admission to critical care areas for determination of NGAL and [TIMP-2] × [IGFBP7] concentrations. The demographic information, comorbidities, clinical, and laboratory data were recorded. The study outcomes were the development of AKI and mortality during hospitalization. Of the 51 individuals that were studied, 25 developed AKI during hospitalization (49%). Of those, 12 had persistent AKI (23.5%). The risk factors for AKI were male gender (HR = 7.57, 95% CI: 1.28–44.8; *p* = 0.026) and [TIMP-2] × [IGFBP7] ≥ 0.2 (ng/mL)^2^/1000 (HR = 7.23, 95% CI: 0.99–52.4; *p* = 0.050). Mortality during hospitalization was significantly higher in the group with AKI than in the group without AKI (*p* = 0.004). Persistent AKI was a risk factor for mortality (HR = 7.42, 95% CI: 1.04–53.04; *p* = 0.046). AKI was frequent in critically ill COVID-19 patients. The combination of [TIMP-2] × [IGFBP7] together with clinical information, were useful for the identification of subclinical AKI in critically ill COVID-19 patients. The role of additional biomarkers and their possible combinations for detection of AKI in ritically ill COVID-19 patients remains to be explored in large clinical trials.

## 1. Introduction

The clinical spectrum resulting from infection with the severe acute respiratory syndrome coronavirus 2 (SARS-CoV-2), ranges from an asymptomatic response or the development of a mild upper respiratory tract infection to critical coronavirus disease 2019 (COVID-19) [1]. The rates of acute kidney injury (AKI) in patients with severe COVID-19 are extremely variable, but the evidence suggests that it likely affects >20% of hospitalized patients and >50% of patients in the intensive care unit (ICU) [2]. Traditionally, AKI diagnosis is based on changes in kidney function such as increased serum creatinine (sCr), which is a low sensitivity method because nearly 50% of glomerular filtration rate (GFR) must be lost before a change in sCr is detectable; and decreased urine output, which lacks specificity since it may be triggered by hypovolemia, without direct damage to the kidney [3]. The recognition of AKI is unacceptably delayed in up to 43% of hospitalized patients [4], leading to loss of therapeutic windows. The elevation of urinary biomarkers of kidney stress in the absence of changes in sCr and urine output has been considered as subclinical AKI, a term for identifying those patients at high risk for AKI [5]. These patients are likely to be the ones who would benefit from the use of biomarkers and early interventions. In this context, the tissue inhibitor of metalloproteinases-2 (TIMP-2) and the insulin-like growth factor binding protein 7 (IGFBP7) have been identified as possible AKI biomarkers, given that both are released following ischemic or inflammatory processes in the kidney, resulting in G1 cell cycle arrest for a short period [6,7,8]. Another biomarker of early AKI is the urinary neutrophil gelatinase-associated lipocalin (NGAL), as intrarenal concentration of this protein is abruptly up-regulated soon after ischemic or nephrotoxic kidney injury [9]. In view of the need to identify the early signs of kidney involvement, this study was aimed at exploring the role of the urinary biomarkers NGAL, TIMP-2, and IGFBP7 for the early detection of AKI in critically ill patients with COVID-19.

## 2. Methods

### 2.1. Study Population

This prospective, longitudinal cohort study was conducted at the National Institute of Respiratory Diseases (INER), the largest third-level institution designated by the Mexican Government for COVID-19 care. The Institutional Review Board approved the study (Approval No C26-20) and written informed consent was obtained from all the participants. We included individuals that were admitted to the ICU with the diagnosis of severe pneumonia that was caused by SARS-CoV-2, who were 18 years of age or older; without AKI when urine sample was collected; and with no history of chronic kidney disease (CKD) as indicated by interrogation of patients about CKD medical history and by an estimated glomerular filtration rate (eGFR) that was greater than 60 mL/min/1.73 m^2^ using the CKD-EPI equation [10]. SARS-CoV-2 severe pneumonia was defined by clinical data of respiratory distress, bilateral alveolar opacities in 2 or more lobes, a ratio of partial arterial oxygen pressure/inspired oxygen fraction (PaO_2_/FiO_2_) < 300 mm Hg, and a positive result for SARS-CoV-2-real-time reverse transcription–polymerase chain reaction (rRT-PCR) assay in a nasopharyngeal swab [11]. The primary outcome was the development of AKI during hospitalization. The secondary outcome was mortality during hospitalization in the group with AKI and the group without AKI. The recorded variables included demographic and anthropometric variables, symptoms, comorbidities, treatments, critical care variables, blood chemistry, blood count, starting and termination dates of invasive mechanical ventilation (IMV), days in hospital, initial mechanical-ventilator settings, use of vasoactive drugs, and outcomes. Pregnant women were not included in the study. Patients with incomplete clinical records were excluded.

### 2.2. Definition of Acute Kidney Injury

AKI staging was based on serum creatinine (sCr) levels. The urine output criterion was not used for diagnosis of AKI since the nursing records were out of reach in COVID-19 areas. The baseline sCr level was defined as the minimum inpatient value during the first 7 days of admission [12]. Diagnosis of AKI was based on the Kidney Disease Improving Global Outcomes (KDIGO) criteria [13]. AKI stage 1 corresponded to an increase in sCr by ≥0.3 mg/dL within 48 h or increase in sCr 1.5 to 1.9 times baseline within the prior 7 days; AKI stage 2 corresponded to an increase in sCr of 2.0–2.9 times baseline; and AKI stage 3 corresponded to an increase in sCr of ≥3 times baseline or the initiation of renal replacement therapy. Persistent AKI was defined by the continuance of AKI by serum creatinine or urine output criteria (as defined by KDIGO) beyond 48 h from AKI onset. Transient AKI was defined by the complete reversal of AKI by KDIGO criteria within 48 h of AKI onset [14].

### 2.3. Biomarker Determinations

The urine samples were collected on admission to critical care areas (day 1). The urine was frozen at −80 °C within the first 30 min after sample collection. The urinary concentrations of TIMP-2 and IGFBP7 were determined using commercially available ELISA kits (Human TIMP-2 Quantikine ELISA Kit, R&D, Minneapolis, Minnesota; Human IGFBP7 ELISA Kit, Abcam, Cambridge, UK) following manual instructions. The ELISA plates were read at O.D. of 450 and the calculations were done according to the signal that was given by the standard curve of each kit. NGAL determinations were done using the NGAL kit (Abbott, Chicago, IL, USA) according to the manual instructions and using the Abbott™ ARCHITECT™ Analyzer.

### 2.4. Statistical Analysis

We performed descriptive statistics including means and standard deviations for normally distributed continuous variables, medians and interquartile ranges for non-parametric distributions, and proportions for categorical variables. The comparisons of individuals who developed AKI during hospitalization vs. those without AKI were made using a chi-squared test for categorical variables and Mann–Whitney U for the continuous variables.

For each biomarker, the area under the receiver-operating characteristics curve (AUC) with 95% confidence intervals was calculated, as well as the sensitivity, specificity, positive predictive value (PPV), negative predictive value (NPV), and the accuracy at 3 different cutoff values using urine samples that were collected upon hospital admission. We considered that the prevalence of AKI for patients with SARS-CoV-2 infection in the ICU was 40% [15]. The cutoffs for each biomarker were selected based on the highest AUC, specificity, and accuracy for prediction of AKI. Combinations of the top biomarkers were also explored. When combinations had no significant added value, the individual biomarkers were preferred. For all analyses, two-sided *p* values ≤0.05 were considered statistically significant. The selected biomarkers and cutoff values were used in the Kaplan–Meier survival analyses for the time to AKI. 

Logistic regression analysis was used to identify the association between the relevant covariates with AKI and mortality. We obtained age-stratified estimates considering 60 years and older as a vulnerable population. The variables were entered into the models when the alpha level of the risk factor was <0.20 in the univariate analysis. Age and gender were entered into the models regardless of the alpha level. All the statistical tests were two-sided, and two-sided *p* values ≤ 0.05 were considered statistically significant. The analysis was conducted using RStudio 1.4.1717.

## 3. Results

### 3.1. Characteristics of Study Population

During the period between May and August 2020, a total of 420 individuals were admitted to the critical areas of the INER. Of those, 69 were negative for SARS-CoV-2 infection; in 44 the infection could not be confirmed; 60 remained in the emergency room due to hospital saturation; and 20 died there. Informed consent could not be obtained for 196 patients. We thus included the 51 patients who provided informed consent for participating in the study (Figure 1). Of those, 30 were male (58.8%); the median age was 53 years (IQR: 40–61); 14 had hypertension (27.5%); 16 had diabetes (31.4%); and 21 were obese 41.2%. Of the 51 individuals that were studied, 25 developed AKI during hospitalization (the AKI group, 49.0%) and 26 did not develop AKI (the non-AKI group, 51.0%). A total of 11 individuals had AKI stage 1 (21.5%); 8 had AKI stage 2 (15.6%); and 6 had AKI stage 3 (11.7%). The baseline characteristics of study population and comparisons between the AKI group vs. the non-AKI group are shown in Table 1.

### 3.2. Performance of Biomarkers as AKI Predictors

Based on the highest AUC, specificity, and accuracy values, the biomarker with best performance for AKI prediction during the whole hospitalization period was NGAL at a cutoff of 45 ng/mL. Considering that most patients developed AKI during the first 7 days at the hospital, we also determined the performance of biomarkers for AKI prediction on day 7 (Table 2). The performance of NGAL was significantly better on day 7 than during the whole hospitalization period (AUC = 0.771 vs. AUC = 0.706; *p* = 0.001). The combination of [TIMP-2] × [IGFBP7] at a cutoff of 0.2 (ng/mL)^2^/1000 were the second best AKI predictors, and the performance of these biomarkers during the whole hospitalization was similar to that of day 7 (AUC= 0.682 vs. AUC= 0.671; *p* = 0.632). 

### 3.3. Time to AKI Was Significantly Shorter in Individuals with Higher Values of Urinary NGAL and [TIMP-2] × [IGFBP7]

As NGAL 45 ng/mL had the highest specificity and accuracy values, we selected this cutoff value for the survival analysis of the time to AKI during hospitalization. Individuals with NGAL ≥ 45 ng/mL were compared vs. those with <45 ng/mL. Urinary NGAL could not be measured in 3 patients. The time to AKI was significantly shorter in individuals with NGAL ≥ 45 ng/mL than in those with <45 ng/mL; *p* = 0.028 (Figure 2).

Similarly, [TIMP-2] × [IGFBP7] 0.2 (ng/mL)^2^/1000 had the highest specificity and accuracy, so this cutoff was selected for the survival analysis of the time to AKI during hospitalization. We found that the time to AKI was significantly shorter in individuals with [TIMP-2] × [IGFBP7] ≥ 0.2 ng/mL than in those with <0.2 (ng/mL)^2^/1000; *p* = 0.017 (Figure 3). 

### 3.4. Elevated Values of Urinary [TIMP-2] × [IGFBP7] were Risk Factors for AKI

The univariate analysis indicated that patients with AKI were older (hazard ratio, HR = 0.94, 95% CI: 0.90–0.99; *p* = 0.034); had a higher frequency of hypertension (HR = 6.02, 95% CI: 1.42–25.40; *p* = 0.014); had levels of [TIMP-2] × [IGFBP7] ≥ 0.2 (ng/mL)^2^/1000 (HR = 5.11, 95% CI: 1.20–21.67; *p* = 0.027); and NGAL ≥ 45 ng/mL (HR = 4.00, 95% CI: 1.15–13.81; *p* = 0.028). After adjusting for possible confounding variables, the multivariate analysis indicated that the risk factors for AKI during the hospitalization period were male gender (HR = 7.57, 95% CI: 1.28–44.8; *p* = 0.026) and [TIMP-2] × [IGFBP7] ≥ 0.2 (ng/mL)^2^/1000 (HR = 7.23, 95% CI: 0.99–52.4; *p* = 0.050) (Table 3). At day 7 of hospitalization, after adjusting by age and sex, we found that only [TIMP-2] × [IGFBP7] ≥ 0.2 (ng/mL)^2^/1000 remained associated with risk for AKI (HR = 5.91, 95% CI: 1.06–32.7; *p* = 0.042). 

### 3.5. Mortality was Higher in Individuals with AKI

We constructed a Kaplan–Meyer curve for mortality comparing the group of 25 patients who developed AKI during follow-up with the group of 26 individuals without AKI. The mortality of individuals who developed AKI at any time during hospitalization was significantly higher than in those who never had AKI, *p* = 0.019 (Figure 4).

Of the 25 individuals with AKI, 13 had transient AKI (25.5 %) and 12 had persistent AKI (23.5%). The CKD-EPI on discharge was similar in individuals with persistent AKI (79.5 mL/min/1.73m^2^, IQR: 17–111) and in those with transient AKI (93 mL/min/1.73m^2^, IQR: 88–104; *p* = 0.53). Logistic regression analysis was used to identify the association between the relevant covariates with mortality. The univariate analysis indicated that mortality was more frequent in patients who were 60 years of age or older (HR = 5.33, 95% CI: 1.23–23.09; *p* = 0.025); and had persistent AKI (HR = 10.83, 95% CI: 1.67–69.91; *p* = 0.012). After adjusting for possible confounding variables, only persistent AKI remained associated with mortality (HR = 7.42, 95% CI: 1.04–53.04; *p* = 0.046) (Table 4).

## 4. Discussion

Upon hospital admission, a large fraction of patients had subclinical signs of kidney dysfunctions that not yet constituted AKI. During the subsequent days, AKI became a common complication in our patients, affecting 49% during hospitalization. This frequency was similar to that which was observed in previous studies, reporting AKI in 50% of the patients with COVID-19 at the ICU [2]. 

We found that [TIMP-2] × [IGFBP7] ≥ 0.2 (ng/mL)^2^/1000 was a risk factor for AKI. In addition, the survival analysis indicated that time to AKI was significantly shorter in individuals with higher [TIMP-2] × [IGFBP7]. To our knowledge, few studies have examined the performance of biomarkers for the prediction of AKI onset in critically ill patients with COVID-19. A small study reported that patients with COVID-19-associated AKI and high levels of [TIMP-2] × [IGFBP7] were more likely to progress to renal replacement therapy than those with AKI but with low [TIMP-2] × [IGFBP7] [16]. Our findings are in line with previous reports, describing elevated levels of [TIMP-2] × [IGFBP7] as predictors of adverse outcomes in various clinical conditions, e.g. death, dialysis, or progression to severe AKI in patients with septic shock [17]; AKI in patients after major surgery [18]; imminent risk of AKI in critically ill patients [7]; and AKI in platinum-treated patients at the ICU [19]. The mechanism that is proposed is that after initial damage, IGFBP7 and TIMP-2 are expressed in tubular cells. IGFBP7 directly increases the expression of p53 and p21, and TIMP-2 stimulates p27 expression, leading to transitory G1 cell cycle arrest, preventing the division of damaged cells [5]. Thus, since the G1 cell cycle arrest is a common response to tubular damage, these biomarkers may better reflect damage regardless of etiology. TIMP-2 is both expressed and secreted preferentially by cells of distal tubule origin, while IGFBP7 is equally expressed across tubule cell types yet preferentially secreted by cells of proximal tubule origin. In human kidney tissue, strong staining of IGFBP7 was observed in the luminal brush border region of a subset of proximal tubule cells, and TIMP-2 stained intracellularly in distal tubules [20]. AKI-induced urinary [TIMP-2] × [IGFBP7] has also been attributed to increased filtration, decreased tubule reabsorption, and proximal tubule cell urinary leakage of both molecules [21].

The combination of [TIMP-2] × [IGFBP7] had the best performance for AKI prediction at values above 0.2 (ng/mL)^2^/1000. This cutoff was based on the overall behavior of the biomarkers in the patients that were studied here. However, different cutoffs for these biomarkers have been reported in other studies, so specific groups of patients may require the identification of optimal cutoff values based on their respective values of AUC, sensitivity, specificity, PPV, NPV, and accuracy. The cutoff values may be affected by the severity of AKI. That is, higher cutoffs may be found in patients with AKI stages 2 and 3; and lower cutoffs may be found in patients with AKI stage 1 or subclinical AKI. Moreover, AKI is a complex syndrome involving a series of complex cellular and molecular pathways, and the different cutoffs may reflect mechanistic differences between the various etiologies of AKI [5]. The pathophysiologic mechanisms of AKI in COVID-19 are thought to be multifactorial including systemic immune and inflammatory responses that are induced by viral infection, systemic tissue hypoxia, reduced renal perfusion, endothelial damage, and direct epithelial infection with SARS- CoV-2 [22]. 

In our cohort, the time to AKI was significantly shorter in individuals with NGAL ≥ 45 ng/mL than in those with <45 ng/mL, but NGAL was not a risk factor for AKI during hospitalization. The fact that performance of NGAL was significantly better on day 7 than during the whole hospitalization period, suggests that NGAL has a narrow predictive time window for AKI, and that may explain why it was not a risk factor for AKI during the whole hospitalization. In addition, NGAL has proved to be less discriminating in the development of septic-associated or adult cardiac-surgery-associated AKI than in other types of AKI, possibly because neutrophils themselves may be a source of NGAL in the setting of systemic inflammation [23].

Contrary to our findings, a recent cohort study found that urinary NGAL > 150 ng/mL predicted the diagnosis, duration, and severity of AKI and acute tubular injury, as well as hospital stay, dialysis, shock, and death in patients with acute COVID-19 [24]. Contrasting results may be explained by the fact that some patients in that study probably had AKI when the urinary samples were collected, while we only included patients without AKI at the time of urine sample collection. Therefore, the median value of NGAL in the AKI group (50.2 ng/mL) and the selected cutoff (45 ng/mL), were far lower in our patients since they had subclinical AKI. In addition, it is unclear if a higher proportion of their patients had AKI stage 2 and stage 3, while most of our patients developed AKI stage 1 on subsequent days. This is relevant because that study also reported a correlation between the urinary NGAL levels and AKI severity. In another recent study, NGAL was also found as an independent risk factor for AKI in patients with COVID-19, but that study also included some patients who already had AKI when the urine samples were collected [25]. Thus, we suggest that in patients with COVID-19, higher NGAL cutoff values seem to be useful in predicting AKI progression but not AKI onset. However, since the number of patients in our study was indeed small, we would not dismiss the possible independent predictive value of NGAL that perhaps could have been revealed by the addition of more patients. Regardless of the selected cutoff values, our findings are in line with a study that reported significantly higher NGAL levels in patients with COVID-19 without evidence of AKI on presentation who subsequently developed AKI stages 1 to 3 within seven days of admission, compared with those who did not develop AKI [26]. In contrast with our findings, the urinary NGAL, but not [TIMP-2] × [IGFBP7], independently predicted AKI in a cohort of decompensated cirrhotic patients, suggesting that different biomarkers should be used in different patient groups [27].

The survival analysis indicated that mortality was more frequent in patients who developed persistent AKI during hospitalization. The concept that time should also be considered in the description of AKI and not only the severity, was demonstrated in a study reporting that the duration of AKI following surgery was independently associated with hospital mortality after adjusting for the severity of illness [28]. Transient AKI may reflect a temporary reduction in renal function without structural damage, whereas persistent AKI would reflect structural tubular damage [29]. Based on these observations, persistent AKI has become a relevant endpoint in subsequent studies and it has consistently been associated with mortality [30].

Since we studied patients with normal kidney function at a baseline, the conclusions of this study may not be applicable for patients with acute-on-chronic renal functional impairment. This is, unfortunately, the disadvantage of renal biomarkers, which provide excellent prediction on evolving AKI in patients with previously intact kidneys but are of limited value in patients with preexisting renal disease. The use of biomarkers has some limitations, and it should be considered that their value for prediction of AKI is limited to patients who are critically ill. When used in patients who are low risk, the false positive rate may increase. When used before an injurious exposure has occurred, the test will not forecast AKI. Similarly, the test might not remain positive for a long time after injury [3]. If positive results are obtained, the test should be interpreted along with other clinical factors and nephrology consultation should be considered. When used properly, biomarker-guided interventions are useful in AKI prevention. This was demonstrated in a clinical trial including high risk patients, defined as urinary [TIMP-2] × [IGFBP7] > 0.3 undergoing cardiac surgery. In that study, the implementation of the KDIGO guidelines, consisting of the optimization of volume status and hemodynamics, avoidance of nephrotoxic drugs, and preventing hyperglycemia, resulted in an absolute risk reduction of 16.6% in the incidence of AKI compared with the standard care [31].

An important limitation of our study was the small sample size. Another study limitation was that patients with incomplete clinical files or those who were transferred to other hospitals due to the scarcity of ICU beds were not included in the study, and this may represent a selection bias. Considering that the standardized definitions of AKI are based on sCr and urine output [32], then inaccessibility to nursing records that are restricted to COVID-19 areas represents an important study limitation because urine output was not used for the diagnosis of AKI, and sCr was not adjusted for fluid-balance. It deserves to be mentioned that both the groups had similar median values of baseline sCr, but we think that the differences between groups might be explained by the fact that in the AKI group, the sCr values were more disperse, interquartile ranges were wider, and individuals were older. The patients with AKI had higher urea levels, but we could not exclude volume depletion in this group. The lack of pre-hospital baseline sCr measurements was also a study limitation because baseline sCr values were an estimation. One additional study limitation was that our study was conducted at a national referral center for respiratory diseases receiving disproportionately more patients with severe COVID-19, and this represents a potential source of referral bias.

## 5. Conclusions

Elevated values of urinary [TIMP-2] × [IGFBP7] were risk factors for AKI and persistent AKI was a risk factor for mortality. These biomarkers, together with clinical information, were useful for the identification of subclinical AKI in critically ill COVID-19 patients. The role of additional biomarkers and their possible combinations for the early detection of AKI in critically ill COVID-19 patients remains to be explored in large clinical trials. Preventable causes of AKI should be reduced. 

## Figures and Tables

**Figure 1 biomolecules-12-00275-f001:**
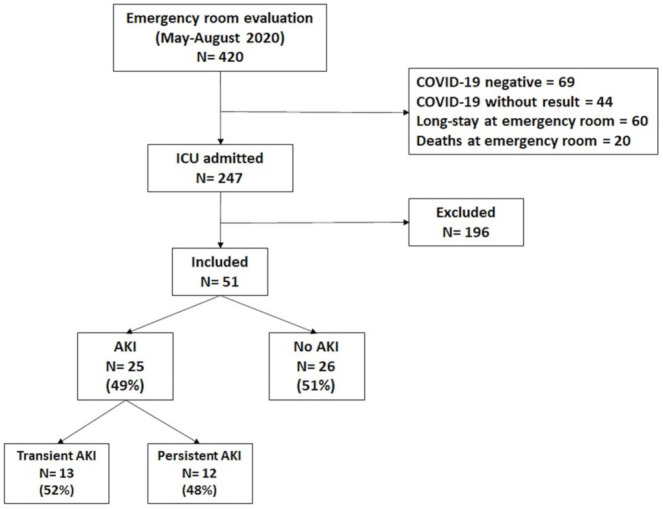
Study diagram. Numbers of individuals that were assessed for eligibility and individuals that were included in the study.

**Figure 2 biomolecules-12-00275-f002:**
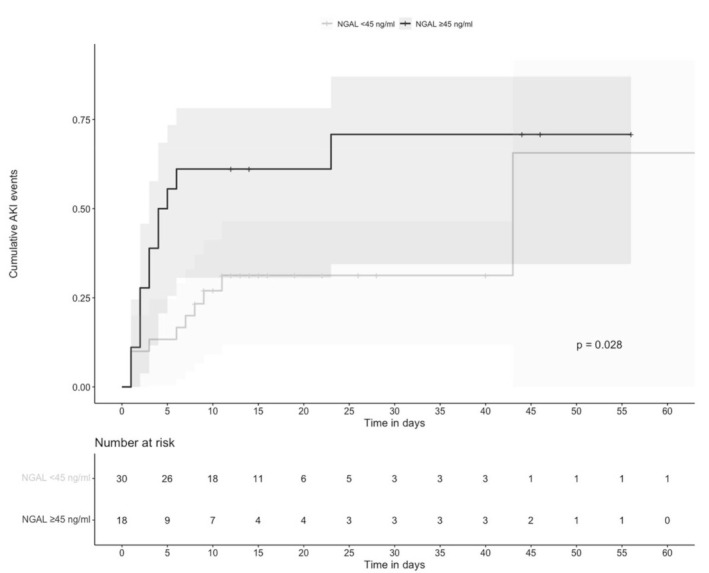
Cumulative AKI events according to urinary NGAL concentrations. AKI in individuals with urinary NGAL > 45 ng/mL (back line) vs. AKI in individuals with urinary NGAL < 45 ng/mL (grey line) during hospitalization (*p* = 0.028). Time 0 corresponds to hospital admission.

**Figure 3 biomolecules-12-00275-f003:**
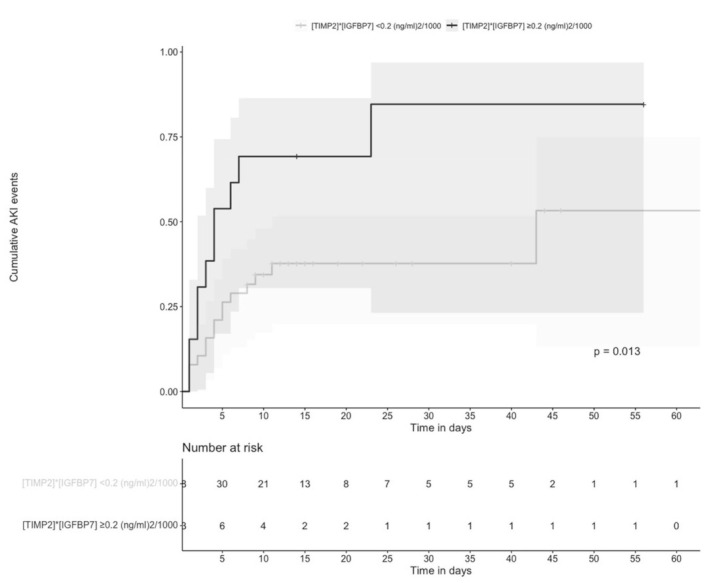
Cumulative AKI events according to urinary [TIMP-2] × [IGFBP7] concentrations. AKI in individuals with urinary [TIMP-2] × [IGFBP7] > 0.2 (ng/mL)^2^/1000 (black line) vs. AKI in individuals with urinary [TIMP-2] × [IGFBP7] < 0.2 (ng/mL)^2^/1000 (grey line) during hospitalization (*p* = 0.013). Time 0 corresponds to hospital admission.

**Figure 4 biomolecules-12-00275-f004:**
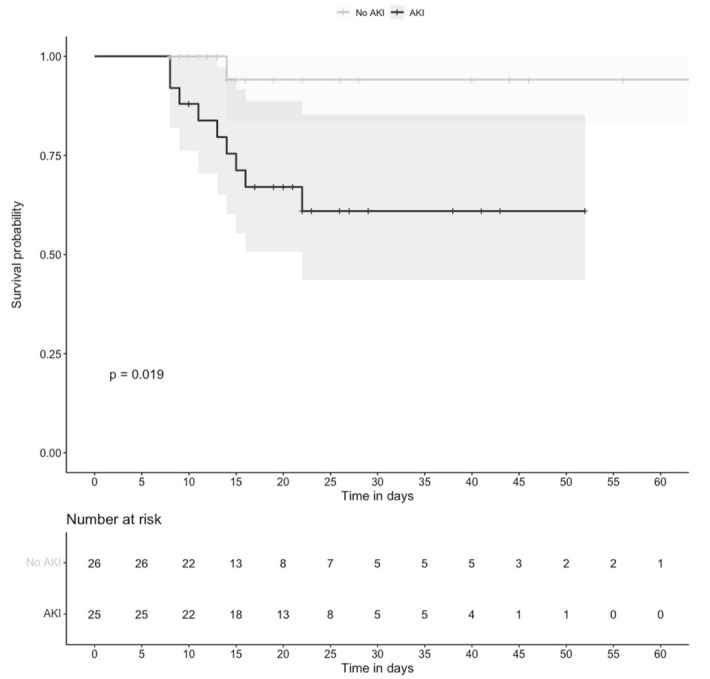
Survival in the AKI group and the non-AKI group. Survival in the AKI group (black line) vs. survival in the non-AKI group (grey line) during hospitalization (*p* = 0.019). Time 0 corresponds to hospital admission.

**Table 1 biomolecules-12-00275-t001:** Baseline characteristics of the study population.

Variable	Overall (*n* = 51)	AKI (*n* = 25)	Non-AKI (*n* = 26)	*p* Value
Age, years *	53 (40–61)	57 (47–65)	49 (37–56)	**0.029**
Male [*n* (%)]	30 (58.8)	18 (72)	12 (46.1)	0.061
BMI, kg/m^2^ *	29.3 (25.9–31.6)	29.4 (26.7–34.1)	29.1 (25.9–30.3)	0.522
**Comorbidities**				
Obesity [*n* (%)]	21 (41.2)	11 (44.0)	10 (38.5)	0.688
Diabetes [*n* (%)]	16 (31.4)	7 (28.0)	9 (34.6)	0.611
Hypertension [*n* (%)]	14 (27.5)	11 (44.0)	3 (11.5)	**0.009**
Two or more comorbidities [*n* (%)]	21 (41.2)	12 (48.0)	9 (34.6)	0.332
**Critical Care Variables**				
IMV [*n* (%)]	32 (62.7)	18 (72.0)	14 (53.8)	0.180
PaO₂/FiO₂ ratio, mmHg *	141 (108–187)	140 (120–174.5)	148 (103–198.5)	0.733
PEEP, cm H₂O *	10 (9.5–14)	10 (8–12)	12 (10–14)	0.125
pH *	7.4 (7.3–7.4)	7.3 (7.3–7.4)	7.4 (7.3–7.4)	0.556
pCO₂, mmHg *	38.0 (32.3–51.9)	46.3 (33.0–52.2)	35.9 (33–50.9)	0.378
SOFA score, points *	4 (2–6)	4 (3–6)	3 (2–6)	0.076
Vasoactive drugs [*n* (%)]	16 (31.4)	8 (32.0)	8 (30.8)	0.925
Inotropic drug [*n* (%)]	2 (3.9)	0 (0.0)	2 (7.7)	0.157
**Treatment**				
Systemic steroids [*n* (%)]	25 (49)	14 (56.0)	11 (42.3)	0.328
Tocilizumab [*n* (%)]	25 (49)	11 (44.0)	14 (53.8)	0.355
Hydroxychloroquine [*n* (%)]	7 (13.7)	4 (16.0)	3 (11.5)	0.643
Lopinavir/Ritonavir [*n* (%)]	12 (23.5)	2 (8.0)	10 (38.5)	**0.010**
Nephrotoxic drugs [*n* (%)]	1 (2)	0 (0.0)	1 (3.8)	0.322
**Renal Function Indicators**				
Serum creatinine, mg/dL, Day 1 *	0.6 (0.5–0.7)	0.6 (0.5–0.8)	0.6 (0.4–0.7)	**0.039**
eGFR, mL/min, Day 1*	112.3 (98.4–121.4)	107.0 (92.5–114.9)	113.3 (108.7–125.1)	**0.026**
**Laboratories**				
Hemoglobin, g/dL, Day 1 *	13.3 (12.6–14.9)	13.1 (12.5–15.0)	13.4 (12.8–14.4)	0.850
Leucocytes, 10×^3^ mm^3^, Day 1 *	8.9 (6.3–13.4)	9.4 (7.8–13.4)	8.3 (5.8–11.6)	0.239
Lymphocytes, 10×^3^ mm^3^, Day 1 *	0.8 (0.6–1.0)	0.7 (0.5–1.0)	0.85 (0.6–1.1)	0.219
Platelets, 10x^3^ mm^3^, Day 1 *	272 (219–329)	283 (228–363)	259 (219–308)	0.323
Blood urea nitrogen, mg/dL, Day 1 *	22 (13–35)	32 (27–45)	13.5 (11–21)	**0.001**
Lactate dehydrogenase, U/mL, Day 1 *	387 (299–557)	421 (327–557)	364.5 (249–541)	0.118
Total bilirrubines, mg/dL, Day 1 *	0.5 (0.4–0.6)	0.5 (0.4–0.6)	0.4 (0.4–0.8)	0.651
Creatine phosphokinase, U/L, Day 1 *	140 (39–443)	175 (97–992)	58 (31–354)	**0.004**
D-dimer, µg/mL, Day 1 *	0.9 (0.4–2.5)	1.1 (0.5–3.7)	0.7 (0.4–1.2)	0.057
C-reactive protein, mg/dL, Day 1 *	16.5 (10.3–27.6)	25.2 (13.6–31.9)	13.2 (10.1–17.7)	**0.030**
Fibrinogen, mg/dL, Day 1 *	733.5 (580.2–821.2)	750 (613.5–805)	685 (587–786)	0.424
Procalcitonin, ng/mL, Day 1 *	0.4 (0.1–0.9)	0.6 (0.2–1.1)	0.1 (0.9–0.5)	**0.043**
Troponin, pg/mL, Day 1 *	5.6 (3.3–37)	12.4 (3.8–40.6)	4.2 (2.1–9.0)	**0.039**
Ferritin, ng/mL, Day 1 *	745.4 (358.3–1883.4)	831.3 (468.2–2512)	636.4 (337.1–1008.7)	0.257
**Urinary Biomarkers**				
NGAL, ng/mL, Day 1 *	39.3 (19.2–98.5)	50.2 (36.4–112.1)	32.9 (13.9–44.7)	**0.015**
TIMP-2, ng/mL, Day 1 *	5.5 (3.0–9.0)	6.3 (4.2–9.1)	5.1 (2.9–7.6)	0.356
IGFBP7, ng/mL/1000, Day 1 *	13.4 (8.2–24.1)	22.1 (9.9–29.0)	11.6 (7.0–17.7)	**0.040**
[TIMP-2] × [IGFBP7], (ng/mL)2/1000, Day 1*	0.08 (0.04–0.24)	0.11 (0.05–0.27)	0.06 (0.02–0.12)	**0.026**
**Outcomes**				
Days in hospital	16 (12–27)	20 (14–27)	14.5 (12–26)	0.304
Days on IMV	13 (10–22.2)	15 (12.5–22)	11 (8.5–21.5)	0.235
Days with symptoms before IMV	10 (6–14)	9 (5.5–12)	11.5 (8–14)	0.178
Mortality [*n* (%)]	10 (19.6)	9 (36.0)	1 (3.8)	**0.004**

AKI, acute kidney injury; BMI, body mass index; IMV, invasive mechanical ventilation; PaO₂/FiO₂, partial arterial oxygen pressure/inspired oxygen fraction; pCO₂, partial pressure of carbon dioxide; PEEP, positive end-expiratory pressure; SOFA, sequential organ failure assessment; eGFR, estimated glomerular filtration rate; Systemic steroids: dexamethasone, methylprednisolone; prednisone; NGAL, neutrophil gelatinase-associated lipocalin; TIMP-2, tissue inhibitor of metalloproteinases 2; IGFBP7, insulin-like growth factor binding protein 7. * Data are expressed as medians (interquartile ranges). Comparisons of the AKI group vs. the non-AKI group were made using chi-squared test for categorical variables and Mann–Whitney U for continuous variables. Bold values denote statistical significance at the *p* ≤ 0.05 level.

**Table 2 biomolecules-12-00275-t002:** Performance of urinary biomarkers for the prediction of acute kidney injury in critically ill COVID-19 patients.

Biomarker	AUC	95% CI	*p*	Cutoff	Sensitivity (%)	Specificity (%)	PPV (%)	NPV (%)	Accuracy (%)
Prediction of AKI during the Whole Period of Hospitalization									
N-Gal (ng/mL)	0.706	0.559–0.854	0.015	40.0	59.1	61.5	50.6	69.3	60.5
				45.0	54.5	76.9	61.1	71.7	67.9
				50.0	50	76.9	59.1	69.7	66.1
[TIMP-2] × [IGFBP7] ((ng/mL)^2^/1000)	0.682	0.535–0.829	0.026	0.1	56.0	69.2	54.8	70.2	63.9
				0.2	40.0	88.4	69.8	68.8	69.1
				0.3	24.0	92.3	67.5	64.5	64.9
**Prediction of AKI on Day 7 of Hospitalization**									
N-Gal (ng/mL)	0.771	0.632–0.910	0.002	40.0	70.5	64.5	57.0	76.6	66.9
				45.0	64.7	77.4	65.6	76.6	72.3
				50.0	64.7	80.6	69.0	77.4	74.2
[TIMP-2] × [IGFBP7] ((ng/mL)^2^/1000)	0.671	0.515–0.827	0.041	0.1	60.0	67.7	55.3	71.7	64.6
				0.2	45.0	87.1	69.9	70.3	70.2
				0.3	25.0	90.3	63.2	64.3	64.1

AKI, acute kidney injury; AUC, area under the receiver-operating characteristics curve; CI, confidence interval; PPV, positive predictive value; NPV, negative predictive value; NGAL; neutrophil gelatinase-associated lipocalin; TIMP-2, tissue inhibitor of metalloproteinases-2; IGFBP7, insulin-like growth factor binding protein 7.

**Table 3 biomolecules-12-00275-t003:** Risk factors for acute kidney injury in critically ill COVID-19 patients.

Variables	Unadjusted HR (95% CI)	*p* Value	Adjusted HR (95% CI)	*p* Value
Age	0.94 (0.90–0.99)	**0.034**	0.95 (0.89–1.01)	0.144
Male	3.0 (0.93–9.61)	0.065	7.57 (1.28–44.8)	**0.026**
Hypertension	6.02 (1.42–25.40)	**0.014**	5.61 (0.84–37.22)	0.074
TIMP-2 × IGFBP7 > 2.0 (ng/mL)^2^/1000	5.11 (1.20–21.67)	**0.027**	7.23 (0.99–52.4)	**0.050**
NGAL > 45	4.00 (1.15–13.81)	**0.028**	1.45 (0.30–6.94)	0.637
CPK	1.00 (0.99–1.00)	0.192	-	-
Procalcitonine > 0.25	2.47 (0.79–7.75)	0.119	0.90 (.204.0)	0.894
Troponine	0.99 (0.98–1.00)	0.274	-	-

HR, hazard ratio; CI, confidence interval; NGAL; neutrophil gelatinase-associated lipocalin; TIMP-2, tissue inhibitor of metalloproteinases 2; IGFBP7, insulin-like growth factor binding protein 7; CPK, creatine phosphokinase. The variables were entered into the model when the alpha level of risk factor was less than 0.15. Age and gender were added into the model regardless of the alpha level. Bold values denote statistical significance at the *p* ≤ 0.05 level.

**Table 4 biomolecules-12-00275-t004:** The risk Factors for mortality in critically ill COVID-19 patients.

Variables	Unadjusted HR (95% CI)	*p* Value	Adjusted HR (95% CI)	*p* Value
Age 60, years	5.33 (1.23–23.09)	**0.025**	3.88 (0.78–19.24)	0.097
Male	3.45 (0.65–18.29)	0.145	2.65 (0.43–16.39)	0.293
Transient AKI	4.33 (0.62–30.24)	0.139	2.47 (0.37–16.38)	0.347
Persistent AKI	10.83 (1.67–69.91)	**0.012**	7.42 (1.04–53.04)	**0.046**

AKI, acute kidney injury; HR, hazard rate. Variables were entered into the model when the alpha level of risk factor was less than 0.15. Age and gender were added into the model regardless of the alpha level. Bold values denote statistical significance at the *p* ≤ 0.05 level.

## Data Availability

All data generated and analyzed during this study were included in a Appendix A.

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
