# Peer review of "Role of Urinary Kidney Stress Biomarkers for Early Recognition of Subclinical Acute Kidney Injury in Critically Ill COVID-19 Patients"

_biomolecules, 2022, doi:10.3390/biom12020275_

Round 1

Reviewer 1 Report

This is a rather small descriptive prospective study, looking at initial urinary [TIMP-2]•[IGFBP7] and NGAL,biomarkers of renal injury in critically ill COVID-19 patients, all with normal kidney function on admission, and their predictive value regarding developing AKI and mortality. They conclude that AKI was common in their series, and that AKI could be predicted by the initial urinary [TIMP-2] • [IGFBP7].

The article is clearly written, and its conclusions are sound. It addresses a small cohort of patients, yet – a well-defined homogenous group with normal baseline kidney function.

Comments

Since HTN was prevalent among AKI patients, I wonder how many of them were on ACE/ARB medications, taking into account the potential impact of these medications in maintaining the balance between the "bad pressor arm" (ACE-Ang II-AT1R) and the "good, depressor arm " (ACE2-Ang-(1-7)-MasR) of the angiotensin family (see Abassi et al. Front Physiol 11: 574753, 2020)

Table 1: provide in legend the statistical method used. I am surprised that the AKI/non-AKI groups supposedly have a statistically significant difference in their baseline kidney function. By eye-test, it doesn't seem so, regarding creatinine and eGFR. Conceivably, the higher urea levels in the AKI group (with comparable creatinines) reflects volume depletion upon admission in patients that subsequently developed AKI, and that should be addressed

Define tubular origin of the urine biomarkers used. To my understanding, while urine NGAL originating from kidney sources is principally generated by medullary thick limbs and collecting ducts, IGFBP7 appears in proximal tubular brush border PT, while TIMP-2 is cytoplasmic in (? cortical ?) distal nephrons (see Am J Physiol Renal Physiol 2017 Feb 1;312(2):F284-F296). You may speculate, based on your findings, on the distribution of evolving tubular injury at least initially, as, for example, is done in Renal Failure 42:836-844, 2020

It should be pointed out that since the study was restricted to patients with normal kidney function at baseline, its conclusions may not be applicable for patients with acute-on-chronic renal functional impairment. This is, unfortunately, the disadvantage of renal biomarkers, which provide excellent prediction on evolving AKI in patients with previously intact kidneys, but are of limited value in patients with preexisting renal disease.

Since numbers are indeed very small, I would not dismiss the possible independent predictive value of NGAL that perhaps could have been revealed by the addition of a few more patients

Reviewer 2 Report

This manuscript by Aparicio et al. looks at clinical factors and urine biomarkers (NGAL and [TIMP2][IGFBP7] as predictors of AKI in an ICU covid population.  It is well written and well organized, and finds that [TIMP2][IGFBP7] predicts AKI after multivariate adjustment.  The impact is held back by the relatively small number of patients due to failure to get informed consent, and other patient recruitment factors, but they are honest about that.  It is a strength that they excluded patients who had AKI on admission, so that their analysis could focus on patients who developed AKI. I have a few other comments as below:

  1. line 124 - "We have considered that the prevalence of AKI for patients with SARS-CoV-2 infection in the ICU was 40%" - I do not understand the point of this sentence, or its use in the statistical analysis.
  2.  Table 1 - how could serum creatinine be statistically different on admission if the value in both  columns was 0.6?
  3. Table 1 - It's interesting that BUN was different between AKI and non-AKI at admission - this suggests the AKI patients were more volume depleted.  This should be commented upon.  Was it used in the multivariate analysis?
  4. Table 1 - [TIMP2][IGFBP7] is listed as 0.1 in AKI and 0.6 in non-AKI - is this backwards?
  5. How does the AUC and test characteristics in this study compare to NGAL and [TIMP2][IGFBP7]in ICU AKI generally?  Does the authors' covid setting make these tests more or less reliable than they are generally?
  6. Line 220-What does "final" CKD-EPI mean - GFR after recovery?  Or what time point?
  7. Line 263 - Actually NGAL was a significant univariate risk factor for AKI, but was no longer a factor after multivariate adjustment.  I recommend being more clear about this.
  8. Line 286-287 - sentence is very redundant, would rewrite.
  9.  Line 305 - Would reword "local saturation" to something like scarcity of ICU beds, or another phrase. 
